# Hepatitis B Vaccination Coverage Rates and Associated Factors: A Community-Based, Cross-Sectional Study Conducted in Beijing, 2019–2020

**DOI:** 10.3390/vaccines9101070

**Published:** 2021-09-24

**Authors:** Yan Liang, Xinxin Bai, Xinyao Liu, Zheng Zhang, Xinghuo Pang, Li Nie, Wuqi Qiu, Wei Zhao, Guangyu Hu

**Affiliations:** 1Chaoyang District Center for Disease Prevention and Control of Beijing, Beijing 100020, China; cyqwjwcdcly@bjchy.gov.cn (Y.L.); cyqwjwcdczz@bjchy.gov.cn (Z.Z.); 2School of Public Health, Hebei Medical University, Shijiazhuang 050011, China; 20202184@stu.hebmu.edu.cn; 3Center for Health Policy and Management, Institute of Medical Information, Chinese Academy of Medical Sciences and Peking Union Medical College, Beijing 100020, China; liu.xinyao@imicams.ac.cn; 4Beijing Center for Disease Prevention and Control, Beijing 100013, China; cyqwjwcdclbk@bjchy.gov.cn (X.P.); bjcdckjb@wjw.beijing.gov.cn (L.N.)

**Keywords:** hepatitis B virus, vaccination, associated factors, China

## Abstract

Hepatitis B vaccination coverage rates are low throughout most populations in China. Factors influencing low coverage rates, including population-specific hepatitis B vaccination barriers, may inform policies that promote vaccination. A cross-sectional survey of residents from 43 communities assessed their vaccination status and identified associated factors via uni- and multivariable logistic regression and subgroup analyses. In total, 11,280 of 36,007 respondents received a hepatitis B vaccine, indicating a 31.33% coverage rate. Multivariable logistic regression revealed non-Beijing (odds ratio (OR) = 0.81; 95% confidence interval (CI): 0.76–0.85) and residents who self-rated their health as very healthy (OR = 0.82; 95% CI: 0.68–0.99) were unlikely to be vaccinated. Farmers (OR = 1.68; 95% CI: 1.51–1.86), commerce and service workers (OR = 1.82; 95% CI, 1.63–2.04), government employees (OR = 1.56; 95% CI: 1.38–1.77), professionals and technicians (OR = 1.85; 95% CI: 1.63–2.09), and students (OR = 1.69; 95% CI: 1.10–2.59) had increased hepatitis B vaccination rates. The multivariable assessment revealed hepatitis B vaccination coverage rates are associated with confirmed or suspected family cases, vaccination unwillingness or uncertainty, and unawareness of its prevention of the hepatitis B virus. Low hepatitis B vaccination coverage rates among Beijing subpopulations highlight the need for improved strategies, including those that target specific populations.

## 1. Introduction

Hepatitis B virus (HBV) is a small, enveloped, primarily hepatotropic DNA virus [1]. Hepatitis B has become an important public health issue, and its disease burden is high globally [2]. In fact, an estimated 296 million people had HBV infections in 2019, and 1.5 million new infections occur each year worldwide [2]. Hepatitis B has resulted in over 0.8 million global deaths, mostly from primary liver cancer, including hepatocellular carcinoma and cirrhosis [3]. China has the world’s largest HBV infection burden, with >83 million individuals positive for the HBV surface antigen (HBsAg) in 2018 [4]. It is estimated that about 0.5% (or 7 million) of the total China population live with liver cirrhosis [5], and HBV has been implicated in the cause of up to 80% of cases of hepatocellular carcinoma (HCC) in China [6]. Liver cancer ranked in the top five of the incidence and mortality of all cancers in China [7]; the mortality of liver cancer is 15.09/10^5^ [7], and HBV infections, a significant mortality source, are the leading cause of liver cancer [6].

HBV can be transmitted by the parenteral route, sexual exposure, and vertical routes. The most common method of transmission is perinatal infection [8]. In high-prevalence areas, such as Southeast Asia and China, perinatal and early childhood horizontal transmission is most common, resulting in high levels of chronicity (95% perinatal, 30% before five years of age) [8]. Immunization is the most effective way to protect people against HBV [9]. To reduce the number of patients infected, the Chinese government integrated hepatitis B vaccination into the national immunization program in 1992, requiring newborns and infants to be vaccinated at their own expense [10]. Since 2002, the hepatitis B vaccine has been included in the national Expanded Program on Immunization (EPI) and provided free-of-charge to newborns [11]. According to two nationwide surveys, the prevalence of HBsAg in those aged 1–60 years decreased from 9.75% in 1992 to 7.18% in 2006, indicating the overall efficacy of hepatitis B vaccination [11]. Data from a national survey in China showed that HBsAg prevalence was only 1% among children younger than five years, but HBsAg prevalence remains high among Chinese adults (8.5–10.5%) [12].

Though hepatitis B vaccination has been shown to help control HBV, vaccination rates vary greatly among populations and various regions. It was reported that hepatitis B vaccination coverage rates in US- and foreign-born women of reproductive age in the United States throughout 2013–2015 were 40.9% and 27.3%, respectively [13]. About three-quarters of clinical laboratory staff underwent hepatitis B vaccination in Sana’a, Yemen [14]. In Flanders, Belgium, hepatitis B vaccination coverage rates in adolescents were <90% [15]. However, few studies have determined current hepatitis B vaccination coverage rates and identified factors associated with vaccination among the Chinese population.

A previous study showed that Beijing’s current vaccination program could significantly reduce the economic burden of HBV-related disease [16]. Here, we aimed to perform a cross-sectional study to explore hepatitis B vaccination coverage rates and related factors among the general population of Beijing for future vaccination strategies.

## 2. Materials and Methods

### 2.1. Study Design and Settings 

We performed a community-based, cross-sectional study between July 2019 and June 2020. We used data from a hepatitis B screening program for the general population of the Chaoyang district in Beijing, which has a large mobile population, as a pilot for the study. The program was used to screen for HBV, determine vaccination history, and offer free hepatitis B vaccination services to the general population. In total, face-to-face surveys were conducted throughout 43 communities within the district. Trained local health workers administered questionnaires, which were used to collect basic information from the study participants, including sociodemographic characteristics, HBV-related information, and hepatitis B vaccination history.

### 2.2. Participants

Respondents were recruited from each considered community. Included respondents were required to be permanent residents of the Chaoyang district, meaning they lived within the Chaoyang district >6 months throughout the past year. Residents aged <1 year were excluded.

### 2.3. Data Collection

Data collected via the HBV screening program were used for current statistical analyses. The following information were included: age, education level, occupation, and household registration. Other characteristics included each individual’s self-rated health status, hepatitis B immunization history, whether or not there was a confirmed or suspected family history of HBV infection, and overall willingness to get vaccinated. The hepatitis B vaccination status of the respondents was determined based on their memory (vaccinated or unvaccinated).

Hepatitis B vaccination history was defined using the following question: “Have you ever been hepatitis B vaccinated before?” Respondents who indicated “yes” and “no” were classified within vaccinated and unvaccinated groups, respectively. Willingness to get vaccinated was defined using the following question: “If the government provides hepatitis B vaccination for free, would you like to be vaccinated?”. Respondents who indicated “yes,” “no,” and “uncertain” were classified within willing, unwilling, and uncertain groups, respectively. The results from this survey indicate the hepatitis B vaccination coverage rate.

### 2.4. Statistical Analyses

Total hepatitis B vaccination coverage rates among respondents with different sociodemographic (age group, marital status, occupation, etc.) and vaccination-related (presence of confirmed or suspected family cases, self-rated health status, willingness to receive hepatitis B vaccination, and whether the individual understood that HBV vaccination is an effective way to protect and control HBV) characteristics were assessed. The X^2^ test was used to compare hepatitis B vaccination coverage rates of different age groups, geographical areas, education levels, occupations, marital statuses, and other related factors. 

Univariate logistic regression analyses were used to assess the association between hepatitis B vaccination coverage rates and the following factors: age group (1–14, 15–24, 25–34, 35–44, 45–54, 55–64, or ≥65 years old), household registration (Beijing or non-Beijing), education level (junior middle school and below, high school graduate, college, postgraduate), occupation (unemployed, farmer, worker, commerce and services, government employee, professionals and technicians, students and retired), marital status (married/cohabiting, single, other), presence of confirmed and suspected cases in the family (yes or no), self-rated health status (very unhealthy, unhealthy, fair, healthy, very healthy), willingness to receive hepatitis B vaccination (yes, no, or uncertain), and the knowledge that hepatitis B vaccination is an effective way to prevent and control HBV (yes or no).

Odds ratios (ORs) and 95% confidence intervals (CIs) were calculated via univariate analyses. Multivariable logistic regression models were then fitted to assess the effects of covariates on univariable associations. The correlations between hepatitis B vaccination coverage rates, age, and education level were analyzed using the Spearman correlation coefficient. In the subgroup analysis, we stratified participants into two groups based on age, those 15–54 years, and those ≥55 years. Individuals aged 15–54 years likely comprised the workforce, and those aged ≥ 55 years comprised the retired population. A subgroup analysis was performed to further assess the association between hepatitis B vaccination coverage ratios of multivariable models and characteristics of populations, including the presence of confirmed or suspected family cases, being willing to get vaccinated, the knowledge that hepatitis B vaccination is an effective way to prevent and control HBV. The strength of associations was measured via ORs and 95% CIs. Data were analyzed using R version 3.6.3, and values of *p* > 0.05 were considered statistically significant.

### 2.5. Quality Control

Specialists guided the statistical design of the study, laboratory testing, logistic services, training plans, and data analysis. Appropriate time was allotted to recall 5% of the total number of respondents at different times to ensure that a large proportion responded. Trained staff conducted all aspects of the local fieldwork. 

## 3. Results

### 3.1. Study Population

In the present study, a total of 36,007 individuals were interviewed. The main characteristics of study participants are reported in Table 1. A total of 62.13% of the respondents were female. The majority of individuals (78.49%) were aged 25–64 years. There were 27,046 (74.86%) individuals included in the study with an education level of high school graduate or lower, and 55.14% were employed. The majority (88.29%) of included individuals were married/cohabiting. More than half (54.77%) of the participants had a self-assessed health status of very healthy or healthy. More than a half (68.81%) of the respondents’ household registration is in Beijing. The majority of respondents (98.84%) do not have HBV family cases, and the majority (82.02%) are willing to get HBV vaccination. More than half (61.94%) have the awareness that HBV vaccination is an effective way to prevent and control HBV. 

### 3.2. Hepatitis B Vaccination Coverage Rates and Associated Factors

Of the 36,007 individuals considered, 11,280 (31.33%) had been vaccinated against hepatitis B (Table 1). Reported hepatitis B vaccination coverage rates decreased as respondent age increased (X^2^ = 3013.69, *p* < 0.001) from 92.16% for participants aged 1–14 years to 20.08% for participants aged ≥65 years. Reported hepatitis B vaccination coverage rates increased as education level increased (X^2^ = 1677.00, *p* < 0.001) from 24.65% for middle school education level to 68.87% for postgraduate education. Hepatitis B vaccination coverage rates of students were significantly elevated (69.62%), while rates in retired (22.02%) and unemployed (22.03%, *p* < 0.001) populations were significantly reduced. Hepatitis B vaccination coverage rates were also significantly elevated in those with HBV family cases (45.35%) and in those with a willingness to undergo hepatitis B vaccination (35.72%, all *p* < 0.001; Table 1).

In the multivariable model, age, household registration, education, occupation, the presence of confirmed and suspected family cases, willingness to undergo hepatitis B vaccination, and knowledge that vaccination is an effective way to prevent and control HBV were associated with hepatitis B vaccination coverage rates (Table 2) compared with age group of 1–14, 15–24 (OR = 0.05; 95% CI, 0.02–0.14), 25–34 (OR = 0.05; 95% CI, 0.02–0.14), 35–44 (OR = 0.04; 95% CI, 0.01–0.13), 45–54 (OR = 0.02; 95% CI, 0.01–0.07), 55–64 (OR = 0.01; 95% CI, 0.00–0.04), 65+ (OR = 0.01; 95% CI, 0.00–0.04). Compared with those whose household registration is in Beijing, those who were not were less likely to undergo hepatitis B vaccination (OR = 0.81; 95% CI: 0.76–0.85). Farmers (OR = 1.68; 95% CI: 1.51–1.86), those employed in commerce and services (OR = 1.82; 95% CI: 1.63–2.04), government employees (OR = 1.56; 95% CI: 1.38–1.77), professionals and technicians (OR = 1.85; 95% CI: 1.63–2.09), and students (OR = 1.69; 95% CI: 1.10–2.59) were more likely to undergo vaccination than those who were unemployed. Further, those who self-rated their health status as very healthy (OR = 0.82; 95% CI: 0.68–0.99) were more unlikely to receive hepatitis B vaccination than those with a self-rated health status of unhealthy.

In the multivariable model, hepatitis B vaccination coverage rates were associated with the presence of confirmed or suspected family cases, unwillingness or uncertainty regarding hepatitis B vaccination, and awareness that HBV vaccination is an effective way to prevent and control HBV, with ORs of 0.57 (0.46–0.70), 0.44 (0.41–0.48) or 0.52 (0.45–0.60) and 0.66 (0.63–0.70), respectively (Table 2). Correlation coefficients of HBV vaccination coverage rates, age, and education were calculated, r = −0.267 and 0.185, respectively, *p* < 0.001.

In the subgroup analysis of the multivariable model, associations between hepatitis B vaccination coverage rates and self-rated health condition, the presence of confirmed or suspected family cases, and the willingness to undergo hepatitis B vaccination remained in both age groups of workers (aged 15–54 years) and retirees (aged ≥ 55 years) (Table 3). In comparison to those with HBV family cases, those without HBV family cases had hepatitis B vaccination coverage rates that were elevated in those aged 15–54 years (60.10% versus 42.72%; OR: 0.51) and those aged ≥ 55 years (30.81% versus 20.49%; OR: 0.64). In comparison with those who were willing to undergo HBV vaccination, those who were unwilling and uncertain in aged 15–54 years were 0.48 times (45.45% versus 24.77%; 95% CI: 0.43–0.54) and 0.61 times (45.45% versus 36.49%; 95% CI: 0.51–0.75) less likely to be vaccinated. In those aged ≥55 years, those unwilling to undergo vaccination were 0.40 times (23.37% versus 10.12%; 95% CI, 0.36–0.45) and 0.44 times (23.37% versus 12.83%; 95% CI, 0.35–0.54) less likely to be vaccinated against hepatitis B. Compared with those who were aware that hepatitis B vaccination is an effective way to prevent and control HBV, those who were unaware had hepatitis B vaccination coverage rates that were elevated both in those aged 15–54 years (47.39% versus 34.74%; OR: 0.69) and in those aged ≥55 years (24.14% versus 15.41%; OR: 0.63).

## 4. Discussion

The current study investigated hepatitis B vaccination coverage rates and associated factors among residents of Beijing’s Chaoyang district, China. To the best of the authors’ knowledge, this is the latest comprehensive study of hepatitis B vaccination coverage rates amongst all age groups that has been performed in Beijing. We found that the overall hepatitis B vaccination coverage rate was 31.33% and that hepatitis B vaccination coverage rates are variable amongst different age groups. Hepatitis B vaccination coverage rates are associated with age, household registration, education level, occupation, the presence of confirmed or suspected family cases, willingness to undergo hepatitis B vaccination, and awareness that hepatitis B vaccination is an effective way to prevent and control HBV.

The overall hepatitis B vaccination coverage rates in our study are similar to a report for a population of respondents from Singapore aged 25–69 years [17] and higher than that of the spouse population of Beijing in 2006 (4.59%) and 2014 (11.16%) [18]. This implies that the policy promoting hepatitis B vaccination in Beijing has achieved encouraging results. 

Findings indicate that unvaccinated individuals tend to be those whose household registration is not in Beijing, are of old age, have low education levels, are unemployed, who self-rate their health status as very healthy, have no confirmed or suspected family cases, are unwilling to undergo hepatitis B vaccination, and are unaware that hepatitis B vaccination is an effective way to prevent and control HBV. Gender did not significantly affect vaccination status (Table 1 and Table 2) and is not associated with the hepatitis B vaccination coverage rates. This finding is different from the results reported by Wibabara et al. [19].

In the present study, the hepatitis B vaccination coverage rates showed a decreasing trend that was statistically significant regarding age, similar to the study of Zhu et al. [20]. The results of this study show a negative correlation between hepatitis B vaccination coverage rates and age. This may be because the Chinese government established the National Hepatitis B Immunization Plan in 1992 for newborns and infants and integrated hepatitis B vaccine into the national expanded immunization program. In addition, the government has provided free vaccination since 2002 [21]. This may be why the hepatitis B vaccination coverage rates of younger people are higher than older people.

In the present study, the hepatitis B vaccination coverage rates showed an increasing trend that was statistically significant with regard to education level, a finding that is in accordance with the results previously reported by both Zhu et al. [20] and Awoke et al. [22]. This may be why an increased awareness of HBV and hepatitis B vaccination and a higher acceptance of vaccination are observed among the highly educated general population [23]. This study shows a positive correlation between hepatitis B vaccination coverage rates and education. According to the results reported by Tan et al., knowledge was one of the top facilitators of hepatitis B vaccination [17]. Public initiatives, including education, should be considered to increase the uptake of HBV vaccination.

There were significant differences between hepatitis B immunization rates of those with different occupations. Students have the highest vaccination coverage rates, followed by professional and technical personnel. The lowest coverage rates observed were among the unemployed workers. This may be related to the higher level of education obtained by professional and technical personnel and the young age of students who received improved health education. Free vaccinations are also given to students when they enter the school in Beijing. 

We observed that the hepatitis B vaccination coverage rates of singles are higher than that of married/cohabiting. This may be because many people are aware that HBV can be transmitted by sexual intercourse, and more singles are more likely to get vaccinated as they may have more sexual partners than married or cohabiting people. Perinatal and early childhood transmissions constitute the major modes of HBV transmission in highly and intermediately endemic areas and maintain HBV endemicity [7]. In 2010, the Ministry of Health recommended screening all pregnant women for anti-HBs, getting a vaccination, and administering hepatitis B immunoglobulin within 24 h of birth for all infants born to hepatitis B surface antigen-positive mothers [21].

We observed differing hepatitis B vaccination coverage rates among different household registration. A reason for this may be that Beijing was among the first few cities to implement a hepatitis B vaccination policy for newborns since 1992 [16], and Beijing has higher medical resources and better health education compared to other cities.

Individuals with confirmed or suspected family cases or knowledge of hepatitis B vaccination had high vaccination rates. Further, those unwilling to undergo hepatitis B vaccination had low vaccination rates in our study. In addition, the associations remained in both the subgroup and multivariable analyses. These findings are in agreement with prior results [24,25]. Therefore, vaccination can be strongly influenced by local policies and by knowledge and attitudes towards vaccines. Interventions should focus on strategies to make people aware that vaccination is the most powerful protection against HBV [26].

It is worth noting that among the community residents, hepatitis B vaccine coverage rates were lowest among those who rated their health status as very healthy in the multivariable logistic regression analysis. In accordance with a previous study reported by Yan et al. [27], this trend was also observed when assessing the influenza vaccination program in Shanghai. The finding that lower vaccination coverage rates may be associated with increased self-rated health status revealed a potential targeted population for future hepatitis B vaccine uptake promotion initiatives. Meanwhile, research on whether the development of new vaccines will affect vaccination coverage rates among different populations is still lacking and is needed in the future [28]. In the coming days, it may be a trend to carry out the policy of free hepatitis B vaccination in areas where conditions are available and even nationwide, and we need the country to invest more financial support to implement this initiative.

Our study had several limitations. First, recall bias may have affected our means of measuring vaccination status because participants may not have accurately recalled their vaccination history, and self-reported answers may lead to bias. Second, this was a single-center, city-based, cross-sectional study, which was vulnerable to selection bias. Third, as a cross-sectional study, findings cannot explain the causal connection between the hepatitis B vaccine coverage rates and associated factors. Future surveys will be needed to examine this relationship.

## 5. Conclusions

This study revealed that about a third of respondents were vaccinated against hepatitis B, implying that the national hepatitis B vaccination coverage among all age groups requires improvement. Residence address, education, occupation, family history, willingness to undergo hepatitis B vaccination, and awareness that hepatitis B vaccination is an effective way to prevent and control HBV were found to be significantly associated with hepatitis B vaccination coverage. Therefore, the results can inform policy makers about populations with low hepatitis B vaccination rates, which should be targeted in the future.

## Figures and Tables

**Table 1 vaccines-09-01070-t001:** Characteristics of respondents in Chaoyang District, Beijing.

Characteristics	N (%)	Vaccination (n)	Rate of Vaccination (%)	X^2^	*p* Value
Gender				1.70	0.192
Male	13,636 (37.87)	4328	31.74		
Female	22,371 (62.13)	6952	31.08		
Age				3013.69	<0.001
1–14	51 (0.14)	47	92.16		
15–24	1098 (3.05)	620	56.47		
25–34	3326 (9.24)	1826	54.9		
35–44	4737 (13.16)	2349	49.59		
45–54	7968 (22.13)	2559	32.12		
55–64	12,228 (33.96)	2554	20.89		
65+	6599 (18.33)	1325	20.08		
Household registration				154.09	<0.001
Beijing	24,776 (68.81)	7255	29.28		
Non-Beijing	11,231 (31.19)	4025	35.84		
Education				1677.00	<0.001
Junior middle school and below	16,173 (44.92)	3986	24.65		
High school graduate	10,780 (29.94)	2945	27.32		
College	8646 (24.01)	4068	47.05		
Postgraduate	408 (1.13)	281	68.87		
Occupation				1656.10	<0.001
Unemployed	3141 (8.72)	692	22.03		
Farmer	7817 (21.71)	2565	32.81		
Worker	710 (1.97)	185	26.06		
Commerce and services	4867 (13.52)	2084	42.82		
Government employee	2992 (8.31)	1224	40.91		
Professionals and technicians	3067 (8.52)	1501	48.94		
Students	158 (0.44)	110	69.62		
Retired	13,255 (36.81)	2919	22.02		
Marital status				644.20	<0.001
Married/cohabiting	31,790 (88.29)	9507	29.91		
Single	2288 (6.35)	1258	54.98		
Other	1929 (5.36)	515	26.7		
Confirmed or suspected family cases				38.07	<0.001
Yes	419 (1.16)	190	45.35		
No	35,588 (98.84)	11090	31.16		
Self-rated health condition				325.24	<0.001
Very unhealthy	701 (1.95)	230	32.81		
Unhealthy	1925 (5.35)	687	35.69		
Fair	8015 (22.26)	2832	35.33		
Healthy	11,706 (32.51)	3883	33.17		
Very healthy	3129 (8.69)	1056	33.75		
Uncertain	10,531 (29.25)	2592	24.61		
Willingness to get HBV Vaccination				791.67	<0.001
Yes	29,534 (82.02)	10191	34.51		
No	5139 (14.27)	797	15.51		
Uncertain	1334 (3.70)	292	21.89		
HBV vaccination is an effective way to prevent and control HBV				524.73	<0.001
Yes	22,302 (61.94)	7966	35.72		
No	13,705 (38.06)	3314	24.18		

**Table 2 vaccines-09-01070-t002:** Logistic regression analysis of factors associated with HBV vaccination.

Factors	Univariable Analysis		Multivariable Analysis	
	OR (95%CI)	*p*	OR (95%CI)	*p*
Gender				
Male	1		1	
Female	0.97 (0.93, 1.02)	0.19	1.03 (0.97, 1.09)	0.37
Age				
1–14	1		1	
15–24	0.11 (0.04, 0.31)	<0.001	0.05 (0.02, 0.14)	<0.001
25–34	0.10 (0.04, 0.29)	<0.001	0.05 (0.02, 0.14)	<0.001
35–44	0.08 (0.03, 0.23)	<0.001	0.04 (0.01, 0.13)	<0.001
45–54	0.04 (0.01, 0.11)	<0.001	0.02 (0.01, 0.07)	<0.001
55–64	0.02 (0.01, 0.06)	<0.001	0.01 (0.00, 0.04)	<0.001
65+	0.02 (0.01, 0.06)	<0.001	0.01 (0.00, 0.04)	<0.001
Household registration				
Beijing	1		1	
Non-Beijing	1.35 (1.29, 1.41)	<0.001	0.81 (0.76, 0.85)	<0.001
Education				
Junior middle school and below	1		1	
High school graduate	1.15 (1.09, 1.21)	<0.001	1.06 (1.00, 1.13)	0.0409
College	2.72 (2.57, 2.87)	<0.001	1.76 (1.65, 1.88)	<0.001
Postgraduate	6.76 (5.47, 8.37)	<0.001	3.71 (2.95, 4.67)	<0.001
Occupation				
Unemployed	1		1	
Farmer	1.73 (1.57, 1.90)	<0.001	1.68 (1.51, 1.86)	<0.001
Worker	1.25 (1.03, 1.50)	0.0211	1.16 (0.96, 1.42)	0.13
Commerce and services	2.65 (2.39, 2.93)	<0.001	1.82 (1.63, 2.04)	<0.001
Government employee	2.45 (2.19, 2.74)	<0.001	1.56 (1.38, 1.77)	<0.001
Professionals and technicians	3.39 (3.04, 3.79)	<0.001	1.85 (1.63, 2.09)	<0.001
Students	8.11 (5.72, 11.50)	<0.001	1.69 (1.10, 2.59)	0.0165
Retired	1.00 (0.91, 1.10)	0.9852	1.32 (1.19, 1.46)	<0.001
Marital status				
Married/cohabiting	1		1	
Single	2.86 (2.63, 3.12)	<0.001	0.92 (0.80, 1.06)	0.2265
Other	0.85 (0.77, 0.95)	0.0024	1.08 (0.97, 1.20)	0.1819
Confirmed or suspected family cases				
Yes	1		1	
No	0.55 (0.45, 0.66)	<0.001	0.57 (0.46, 0.70)	<0.001
Self-rated health condition				
Very unhealthy	1		1	
Unhealthy	1.14 (0.95, 1.36)	0.1713	1.02 (0.84, 1.25)	0.8108
Fair	1.12 (0.95, 1.32)	0.1797	1.12 (0.94, 1.33)	0.2199
Healthy	1.02 (0.86, 1.20)	0.8438	0.95 (0.80, 1.13)	0.5444
Very healthy	1.04 (0.88, 1.24)	0.6344	0.82 (0.68, 0.99)	0.043
Uncertain	0.67 (0.57, 0.79)	<0.001	0.55 (0.46, 0.65)	<0.001
Willingness to get HBV Vaccination				
Yes	1		1	
No	0.35 (0.32, 0.38)	<0.001	0.44 (0.41, 0.48)	<0.001
Uncertain	0.53 (0.47, 0.61)	<0.001	0.52 (0.45, 0.60)	<0.001
HBV vaccination is an effective way to prevent and control HBV				
Yes	1		1	
No	0.57 (0.55, 0.60)	<0.001	0.66 (0.63, 0.70)	<0.001

**Table 3 vaccines-09-01070-t003:** Associations between related factors and HBV vaccination coverage rates stratified by age group *.

Factors	N	15–54			N	≥55		
		HBV Vaccination(%)	OR(95%CI)	*p*		HBV Vaccination(%)	OR(95%CI)	*p*
Self-rated health condition								
Very unhealthy	349	150 (42.98)	1		352	80 (22.73)	1	
Unhealthy	900	440 (48.89)	1.06 (0.82, 1.38)	0.653	1025	247 (24.10)	0.97 (0.72, 1.31)	0.798
Fair	3279	1688 (51.48)	1.26 (0.99, 1.59)	0.057	4735	1143 (24.14)	0.98 (0.76, 1.29)	0.212
Healthy	5399	2470 (45.75)	1.03 (0.82, 1.29)	0.821	6303	1409 (22.35)	0.86 (0.66, 1.13)	0.558
Very healthy	1772	788 (44.47)	0.92 (0.72, 1.17)	0.502	1350	262 (19.41)	0.71 (0.53, 0.96)	0.042
Uncertain	5430	1818 (33.48)	0.57 (0.45, 0.72)	<0.001	5062	738 (14.58)	0.55 (0.42, 0.72)	<0.001
Confirmed or suspected family cases								
Yes	208	125 (60.10)	1		211	65 (30.81)	1	
No	16,921	7229 (42.72)	0.51 (0.38, 0.69)	<0.001	18,616	3814 (20.49)	0.64 (0.47, 0.87)	0.004
Willingness to get HBV Vaccination								
Yes	14,757	6707 (45.45)	1		14,731	3442 (23.37)	1	
No	1865	462 (24.77)	0.48 (0.43, 0.54)	<0.001	3270	331 (10.12)	0.40 (0.36, 0.45)	<0.001
Uncertain	507	185 (36.49)	0.61 (0.51, 0.75)	<0.001	826	106 (12.83)	0.44 (0.35, 0.54)	<0.001
HBV vaccination is an effective way to prevent and control HBV								
Yes	11,090	5256 (47.39)	1		11,207	2705 (24.14)	1	
No	6039	2098 (34.74)	0.69 (0.65, 0.74)	<0.001	7620	1174 (15.41)	0.63 (0.58, 0.68)	<0.001

* Adjust for: age, education, occupation, marital status, household registration.

## Data Availability

The data presented in this study are available upon request from the corresponding author. The data are not publicly available due to privacy.

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
