# Peer review of "Hepatitis B Vaccination Coverage Rates and Associated Factors: A Community-Based, Cross-Sectional Study Conducted in Beijing, 2019–2020"

_vaccines, 2021, doi:10.3390/vaccines9101070_

Round 1
Reviewer 1 Report
General Comments:
The authors performed a community-based cross-sectional study July 2019 to June 2020 (1 year) in a district of Bejing about a genral screen for HBV including th vaccination history and HBV-related sociodemographic information by questionnare and interviews. At the same time they offered vaccination.
The authors intended to get an assessment of HBV vaccination coverage rate in dependency of age (measured in different decades), household registration, education levels, occupatients, marital status, cases of hepatitis in the family, self-rated health status, willingness to get HBV vaccination, and the knowledge about the benefit of HBV vaccination. 360007 individuals were interviewed. 31.39% had been vaccinated before.
It seems that only interviews (oral or written by questionnares) were taken, no blood tests. The authors created a profile oft he HBV vaccination coverage rate of different sociodemographic population groups. This has to be considered together with the background that in China hepatitis B is endemic with all oft he associated health problems and a high mortality. It would be interesting to know something about the rate of natural immunization and infection. Statistics were done by uni- and multivariate analysis.
The results were as expected. Younger age, high level of education and school training and professional work as well as the knowledge about the disease are associated with a high vaccination rate. A background of actual references regarding hepatits B oft he last years are given.
The effort has to be acknowledged to get information where to intensify the informing oft he population about the disease and to advertise vaccination. Thus the results oft he study could be a starting point for a general and important health care.
Author Response
Point 1: It seems that only interviews (oral or written by questionnaires) were taken, no blood tests.
Response 1: Hepatitis B vaccine is an effective strategy to prevent hepatitis B virus infection. However, according to Farahnaz et al, there is still 11.1% of inoculator vaccinated presented the status of no response [1]. It is inaccurate to use the positive rate of anti-HBs as a substitute for hepatitis B vaccination rates. That’s why our study did not analyze the serological results of the population.
Point 2: The authors created a profile of the HBV vaccination coverage rate of different sociodemographic population groups. This has to be considered together with the background that in China hepatitis B is endemic with all of the associated health problems and a high mortality. It would be interesting to know something about the rate of natural immunization and infection.
Response 2: We revised the section as follows: “China has the world’s largest HBV infection burden, with>83 million individuals positive for HBV surface antigen (HBsAg) in 2018 [2]. It is estimated that about 0.5% (or 7 million) of the total China population live with liver cirrhosis [3], and HBV has been implicated in the cause of up to 80% of cases of hepatocellular carcinoma (HCC) in China [4]. Liver cancer incidence and mortality ranked in the top 5 of all cancers in China [5], the mortality of liver cancer is 15.09/105[5], and HBV infections, which are a significant source of mortality, are the leading cause of liver cancer [6].”
We are grateful for the reviewers’ in-depth analysis and professional opinion. Most individuals with chronic HBV infection acquired the virus through vertical transmission, highlighting the importance of active and passive immunization for HBV. Without immunoprophylaxis with hepatitis B immunoglobulin(HBIG) and administration of the first HBV vaccine series before newborn discharge, perinatal HBV infection leads to chronic hepatitis in more than 90% of infection children, increasing mortality risk caused by cirrhosis and HCC later in adulthood. Adults may present with acute hepatitis but chronicity is rare in the population [7]. According to the study of Mariko et al., the overall rate of HBsAg seroclearance was 1.75% annually[8].
References:
- Zhang, S., et al., Cancer incidence and mortality in China, 2015. Journal of the National Cancer Center, 2021. 1(1): p. 2-11.
- Liu, J., et al., Countdown to 2030: eliminating hepatitis B disease, China. Bull World Health Organ, 2019. 97(3): p. 230-238.
- Xiao, J., et al, Global liver disease burdens and research trends: Analysis from a Chinese perspective. Journal of hepatology, 2019. 71(1): p. 212-221.
- Song, C., et al., Associations Between Hepatitis B Virus Infection and Risk of All Cancer Types. JAMA Network Open, 2019. 2(6): p. e195718.
- Zhang, S., et al., Cancer incidence and mortality in China, 2015. Journal of the National Cancer Center, 2021. 1(1): p. 2-11.
- Song, C., et al., Associations Between Hepatitis B Virus Infection and Risk of All Cancer Types. JAMA network open, 2019. 2(6): p. e195718-e195718.
- Mysore, K.R. and D.H. Leung, Hepatitis B and C. Clinics in Liver Disease, 2018. 22(4): p. 703-722.
- Kobayashi, M., et al., Seroclearance rate of hepatitis B surface antigen in 2,112 patients with chronic hepatitis in Japan during long-term follow-up. Journal of Gastroenterology, 2014. 49(3): p. 538-546.
Reviewer 2 Report
The article from Xinxin Bai et al. reflect on the correlation between vaccination rate in the Chinese population against hepatitis B and various socio-economic factors like profession, education status and age.
They describe the relative rate of vaccination to give policy makers the possibility to target specific groups whose rate of vaccination specifically low.
I think they succeed with this intention by identifying in particular older people as a group with a low vaccination rate.
Author Response
Point 1: The article from Xinxin Bai et al. reflect on the correlation between vaccination rate in the Chinese population against hepatitis B and various socio-economic factors like profession, education status and age.
They describe the relative rate of vaccination to give policy makers the possibility to target specific groups whose rate of vaccination specifically low.
I think they succeed with this intention by identifying in particular older people as a group with a low vaccination rate.
Response 1: Thank you a lot for your hard work and your recognition of our work. We will work harder to do better in scientific research in the futher.
Reviewer 3 Report
This is an interesting paper.
1) This paper needs to be thoroughly edited by a good English editor. Too many grammatical errors and incorrect usages. This is, however, understandable as English is not the first language of the authors.
2) It must be stated that HBV is a DNA virus
3)There is no mention of HBV modes of transmission. I think that this is a serious mistake in the paper has much to do with its spread:
https://www.ncbi.nlm.nih.gov/pmc/articles/PMC4292086/
For example, I noticed that in the data from the paper that singles tend to be more vaccinated than married people. Apparently, the data seem ti be saying many people may be aware that HBV can be transmitted via sexual intercourse. As a result, more singles are likelier to get vaccinated as they may have many sexual partners, whereas married people do not feel the need for the vaccination as they believe that they have a monogamous relationship.
4) According to Burns et al (above), "In high-prevalence areas, such as Southeast Asia and China, perinatal and early childhood horizontal transmission is most common". Do you have any more data pertaining to married women who wants to get pregnant? Are they more vaccinated? Apparently, Burns et al is saying that it is more important in China and SE Asian countries that ladies who plan on getting pregnant should get vaccinated. What does your statistics has to offer for this?
5) Since I can see that there is a negative correlation between vaccination rate and age but there is also a positive correlation between vaccination rate and educational level, the authors should try to calculate the r (correlation) or r2 ( coefficient of determination) for both. The reader will then know the strength of the correlations. For example, if there is very strong correlation between education level and vaccination rates, perhaps people need to be re-educated on the need of vaccination.
5) Your data show differences between vaccination rates in Beijing and other provinces. What are the "other" provinces. Are the provinces mainly rural or cities?
6) Burns et al mentioned that adults older than 19 tend to be more infected by HBV, perhaps because of their sexual activities. How can your data relate to this?
Author Response
Point 1: This paper needs to be thoroughly edited by a good English editor. Too many grammatical errors and incorrect usages. This is, however, understandable as English is not the first language of the authors.
Response 1: Thank you for your comments, we have already modified the language of the article.
Point 2: It must be stated that HBV is a DNA virus
Response 2: We revised the section as follows: “Hepatitis B virus (HBV) is a small, enveloped, primarily hepatotropic DNA virus.”
Point 3: There is no mention of HBV modes of transmission. I think that this is a serious mistake in the paper has much to do with its spread:
Response 3: We revised the section as follows:
Introduction - “HBV can be transmitted by parenteral, sexual exposure, and vertical routes. The most common method of transmission is perinatal infection [7]. In high-prevalence areas, such as Southeast Asin and China, perinatal and early childhood horizontal transmission is most common, resulting in high levels of chronicity (95% perinatal, 30% before five years of age) [7].”
Discussion - “We observed that the hepatitis B vaccination coverage rates of the singles are higher than that of married/cohabiting. This may be because many people are aware that HBV can be transmitted by sexual intercourse, and more singles are more likely to get vaccinated as they may have more sexual partners than married or cohabiting people. Perinatal and early childhood transmissions constitute the major modes of HBV transmission in highly and intermediately endemic areas and maintain HBV endemicity [7]. In 2010 the Ministry of Health recommended screening all pregnant women for anti-HBs, getting a vaccination, and administering hepatitis B immunoglobulin administration within 24 hours of birth for all infants born to hepatitis B surface antigen-positive mothers[22].”
Point 4: According to Burns et al (above), "In high-prevalence areas, such as Southeast Asia and China, perinatal and early childhood horizontal transmission is most common". Do you have any more data pertaining to married women who wants to get pregnant? Are they more vaccinated? Apparently, Burns et al is saying that it is more important in China and SE Asian countries that ladies who plan on getting pregnant should get vaccinated. What does your statistics has to offer for this?
Response 4: We are sorry that we didn’t have data on married women who wants to get pregnant. We will strengthen this research in the future.
Point 5: Since I can see that there is a negative correlation between vaccination rate and age but there is also a positive correlation between vaccination rate and educational level, the authors should try to calculate the r (correlation) or r2 (coefficient of determination) for both. The reader will then know the strength of the correlations. For example, if there is very strong correlation between education level and vaccination rates, perhaps people need to be re-educated on the need of vaccination.
Response 5: We revised the section as follows:
Materials and Methods - “The correlations between hepatitis B vaccination coverage rates, age and education level were analyzed using the Spearman correlation coefficient.”
Results - “Correlation coefficients of HBV vaccination coverage rates, age, and education were calculated, r=-0.267 and 0.185, respectively, P<0.001.
Discussion - “In the present study, the hepatitis B vaccination coverage rates showed a decreasing trend that was statistically significant regarding age, similar to the study of Zhu et al [21]. The results of this study show a negative correlation between hepatitis B vaccination coverage rates and age.”“This study shows a positive correlation between hepatitis B vaccination coverage rates and education.”
Point 6: Your data show differences between vaccination rates in Beijing and other provinces. What are the "other" provinces. Are the provinces mainly rural or cities?
Response 6: We have replaced the other provinces in China with non-Beijing. Other provinces mean the place outside of Beijing. As the capital of China, Beijing has a relative developed economic and medical care than non-Beijing, and Beijing was among the first few cities to implement a hepatitis B vaccination policy for newborns since 1992(16). We want to explore the difference of hepatitis B vaccination coverages rates of household registration of Beijing and non-Beijing.
Point 7: Burns et al mentioned that adults older than 19 tend to be more infected by HBV, perhaps because of their sexual activities. How can your data relate to this?
Response 7: Thank you to the reviewer for our valuable comments. In our study, we observed a decreasing trend in vaccination rates with age and we didn’t analyze the rate of hepatitis B infection in populations of different ages and their sexual activities in this study, maybe we will do further research on this area.
Round 2
Reviewer 3 Report
"The overall hepatitis B vaccination coverage rates in our study are similar from a 223 report for..." should be
"The overall hepatitis B vaccination coverage rates in our study are similar TO a 223 report for..."
Author Response
Point 1:"The overall hepatitis B vaccination coverage rates in our study are similar from a 223 report for..." should be
"The overall hepatitis B vaccination coverage rates in our study are similar TO a 223 report for..."
Response 1:Thanks for your preciseness in pointing out the syntax error,which we have corrected it.
We revised the section as follows:"The overall hepatitis B vaccination coverage rates in our study are similar to a report for a populatioin of respondents from Singgapore aged 25-69 years..."